# Cocaine and Amphetamine Regulated Transcript (CART) Expression Changes in the Stomach Wall Affected by Experimentally Induced Gastric Ulcerations

**DOI:** 10.3390/ijms22147437

**Published:** 2021-07-12

**Authors:** Michal Zalecki, Adrianna Plywacz, Hanna Antushevich, Amelia Franke-Radowiecka

**Affiliations:** 1Department of Animal Anatomy, Faculty of Veterinary Medicine, University of Warmia and Mazury in Olsztyn, ul. Oczapowskiego 13, 10-719 Olsztyn, Poland; aplywacz@gmail.com (A.P.); ameliaf@uwm.edu.pl (A.F.-R.); 2Department of Genetic Engineering, The Kielanowski Institute of Animal Physiology and Nutrition, Polish Academy of Sciences, ul. Instytucka 3, 05-110 Jabłonna, Poland; a.antuszewicz@ifzz.pl

**Keywords:** cocaine- and amphetamine-regulated transcript, CART, enteric nervous system, stomach, ulcer, Q-PCR, immunohistochemistry

## Abstract

Cocaine- and amphetamine-regulated transcript (CART) is a peptide suggested to play a role in gastrointestinal tract tissue reaction to pathology. Gastric ulceration is a common disorder affecting huge number of people, and additionally, it contributes to the loss of pig livestock production. Importantly, ulceration as a focal disruption affecting deeper layers of the stomach wall differs from other gastrointestinal pathologies and should be studied individually. The pig’s gastrointestinal tract, due to its many similarities to the human counterpart, provides a valuable experimental model for studying digestive system pathologies. To date, the role of CART in gastric ulceration and the expression of the gene encoding CART in porcine gastrointestinal tube are completely unknown. Therefore, we aimed to verify the changes in the CART expression by Q-PCR (gene encoding CART in the tissue) and double immunofluorescence staining combined with confocal microscopy (CART immunofluorescence in enteric nervous system) in the porcine stomach tissues adjacent to gastric ulcerations. Surprisingly, we found that gastric ulcer caused a significant decrease in the expression of CART-encoding gene and huge reduction in the percentage of CART-immunofluorescent myenteric perikarya and neuronal fibers located within the circular muscle layer. Our results indicate a unique CART-dependent gastric response to ulcer disease.

## 1. Introduction

Cocaine- and amphetamine-regulated transcript (CART), a relatively recently discovered peptide, is being recognized as a substance playing an important role in the peripheral nervous system and is known to be expressed in nerves supplying various organs, especially the digestive tract. Importantly, its role in tissue reaction to pathological circumstances has been also observed, indicating its potential role in adaptive changes of tissues to pathology. Neuronal tissue is known to possess a special ability to respond to a pathological factor, which is referred to as ‘neuronal plasticity’. Stomach, as a part of the digestive tract, is widely innervated by extrinsic and intrinsic nerves, and the later form a complicated and highly autonomic enteric nervous system [1]. Multiple experiments have revealed the reaction of enteric neurons to pathology, which was manifested by changes in expression of different neuropeptides by intrinsic nerve cells [2]. It should be stressed, that gastrointestinal tissue and nerve reactions are highly specific and strongly dependent on the influencing factor and the part of the intestinal tube [1,2]. Importantly, CART-dependent reaction to pathology was studied in the intestinal tissues [3,4,5,6,7,8,9,10], while only occasional studies dealt with gastric tissues. The only available studies verified the changes in CART-immunoreactive stomach neurons during experimentally induced diabetes mellitus type I [11], intoxication with T2 toxin [12], acrylamide [13], and bisphenol a [14] supplementation, which are all multi-system pathologies affecting variety of tissues. Thus, such results should be related to complex multi-system reactions, but not the pathologies typically associated with the stomach. Our previous experiments have clearly shown that gastric ulcers strongly affected the enteric nervous system in the stomach wall [15,16,17,18] and these complex changes differed in several respects from other gastrointestinal pathologies. Importantly, ulcers in the distal part of the stomach (pyloric antrum) affect the gastric emptying [19,20] leading to maldigestion, malabsorption, and malnutrition. These specific symptoms were even referred to as “pyloric syndrome complex” [21].

Since gastric ulcerations affect a huge number of people around the world and, in addition, significantly contribute to the loss of pig livestock production [22,23], they represent a serious problem in human and animal medicine. Moreover, the pig—as an omnivorous animal with many embryological, histological, and functional similarities to humans—is considered as the best animal model for biomedical studies on gastrointestinal tract [24,25,26]. Therefore, the basic science knowledge on substances involved in the regulation of porcine gastric ulcer disease is highly valuable for the future experiments. Thus, in the present experiment we aimed to verify by double immunofluorescence and confocal microscopy whether CART is involved in the stomach enteric nerve’s reaction to the gastric antrum ulceration in tissues adjacent to the pathological focus. Importantly, tissue reaction to pathology may be additionally manifested by changes in the expression of genes encoding various substances, and such genetic reaction is of key importance in tissue adaptive changes. The first demonstration of CART mRNA in gastrointestinal tissues was difficult to obtain even in laboratory rodents [27,28]. To date, no study has described the expression of CART encoding gene in the swine gastrointestinal tract tissues, and neither experiment verified its expression changes under pathological processes occurring in the pig gastrointestinal tube. Therefore, our second scientific aim was to demonstrate the expression of gene encoding CART in the porcine stomach tissues and verify the changes in its expression under stomach ulcerations using Q-PCR technique (in tissues strictly corresponding with samples tested by immunofluorescence).

## 2. Results

### 2.1. CART-Immunofluorescence in the Structures of Enteric Nervous System

Microscopic analysis revealed that in all studied animals CART-immunofluorescence was observed mainly in the myenteric perikarya (Figure 1A–D) and nerve fibers located within muscular layer (Figure 1E,G), while only occasional irregularly arranged submucosal perikarya were noticed in a few individuals of both animal groups (Figure 2A–D) and no difference in their occurrence and number was noticed.

#### 2.1.1. CART-Immunoreactive Myenteric Plexus Perikarya

CART-immunoreactive perikarya constituted 39.98 ± 1.24% of myenteric cells in control animals, while in ulcer pigs this number decreased to 34.9 ± 0.48%, and observed difference was statistically significant (Figure 3). In both animal groups most of the PGP 9.5/CART-immunofluorescent myenteric perikarya were oval or multipolar in shape, however they differed in dimensions and fluorescence intensity (Figure 1A–D and Figure 4A–D).

Confocal software measurements revealed that in the control animals the average CART-immunofluorescent myenteric perikaryon measured 12.11 ± 0.28 × 22.35 ± 0.44 µm, while in ulcer pig 13.43 ± 0.31 × 24.04 ± 0.53 µm, and observed difference was statistically significant (Figure 5). However, larger and smaller perikarya were observed in both animal groups (Figure 4E–H).

ImageJ software analysis revealed that in the control pigs average CART-immunofluorescence intensity of myenteric perikarya amounted to 77.25 ± 1.88 units, while in ulcer animals it significantly increased, up to 89.11 ± 3.1 units (Figure 6).

#### 2.1.2. CART-Immunoreactive Nerve Fibers

CART-immunoreactive nerve fibers run mostly parallel to muscle fibers. Most numerous CART-immunoreactive nerve fibers were observed within the circular muscle layer. They were characterized by medium to strong immunofluorescence and penetrated the full thickness of the layer (Figure 1E,G). ImageJ software analysis of the surface area occupied by CART- fluorescent fibers located in the circular muscular layer (Figure 1F,H) revealed its statistically significant decrease from 1.14 ± 0.08% in control animals to 0.83 ± 0.06% in experimental pigs (Figure 7).

In the longitudinal muscle layer, the section thickness of which showed significant differences between different parts of the same specimen, examined nerve fibers were evenly scattered within the layer and no differences in their occurrence were noticed between control and experimental animals (Figure 8A,B).

In both animal groups singular, highly CART-immunoreactive neuronal fibers were observed in the submucosa and lamina muscularis mucosae (Figure 8C–F), which run mostly parallel to the muscularis mucosae fibers. Neuronal fibers of these layers frequently bordered or encircled submucosal, CART-negative, perikarya (Figure 8G,H). Occasionally bundles of prominent fibers were additionally noticed (Figure 8E,F) in both animal groups. No CART-immunofluorescent nerve fibers penetrated mucosa in all animals.

### 2.2. Expression of mRNA Encoding CART in the Stomach Wall

Results of the Q-PCR revealed the expression of gene encoding CART in the stomach tissues of the control and experimental pigs. Statistical analysis of its relative expression (in relation to GAPDH as a housekeeping gene) revealed its significant decrease in stomachs of ulcer animals, in relation to control pigs (Figure 9).

## 3. Discussion

Our experiment has revealed for the first time that gastric ulceration influenced the CART-dependent intrinsic nerve and tissue reaction. Surprisingly, in the stomach wall adjacent to the ulcer focus the number of CART-immunoreactive myenteric perikarya and the expression of CART-IF in nerve fibers penetrating the muscular circle layer were significantly reduced. Moreover, the dimensions and intensity of CART-fluorescence was changed in myenteric perikarya of ulcer animals. In line, the expression of firstly demonstrated CART encoding mRNA was reduced in studied tissues of ulcer pigs. Perikarya of the submucosal ganglia and neuronal fibers located in the longitudinal muscle layer and submucosa were not engaged in CART reaction to stomach ulcerations.

CART-immunoreactive nerve structures are widely distributed within the enteric nervous system of different species [29]. Interestingly, its expression hugely differs between the part of the digestive tract and species examined [29,30]. Stomach is a specific reservoir of food, in which mechanic mixing coupled with acidic digestion occurs. Moreover, pyloric sphincter in cooperation with pyloric antrum precisely adjusts the flow of chyme to the following parts of the digestive tract (intestines), hugely influencing the proper digestion and absorption of nutrients in the intestines. Therefore, the precise regulation of the stomach motility and gastric juice secretion are of particular importance for proper nutrition and gastrointestinal homeostasis. Until this point, data [11,31,32,33,34,35] as well as obtained neuroanatomical results consistently, suggest that—in healthy individuals CART—is mainly engaged in gastric motility regulation, as depicted by its presence in myenteric perikarya and nerve fibers supplying the muscular layer. Interestingly, the percentage of CART-immunoreactive myenteric perikarya (in relation to all myenteric neurons) varied depending on the stomach compartment studied and the researcher: in the stomach antrum such cells constituted as many as 40% [31]—50% [11]; in the gastric corpus 18% [31,32]—30% [11] until 46% [12]; and in the pylorus these perikarya accounted for 14% [11] to over 30% [31] of all myenteric neurons. Above described ‘site-dependent’ discrepancies indicate specific engagement of CART in motor activity regulation in each stomach compartment, while variations regarding the same stomach compartment presented by different authors may result from the precise location of the sample in a given part of the stomach. Porcine stomach is a huge anatomical structure, thus 2–3 cm^2^ specimens sampled for analysis by most of these authors may include unidentical sample localizations, although obtained from the same gastric compartment. In our studies we analyzed the entire transverse cross-section of the pyloric antrum wall (from small to large curvature) in the precisely defined location to obtain the most reliable data. Our results depicted that in the pyloric antrum CART-immunoreactive myenteric perikarya constitute about 40% of all myenteric neurons, strongly suggesting their involvement in the motility regulation in this stomach compartment.

CART-IF neuronal fibers were numerously observed in the muscular layer of the stomach antrum [11,31], corpus [11,12,31,32], pylorus [11,31], and the pyloric antrum, as depicted by our present study, which strongly supports the hypothesis about the influence of CART on the gastric smooth muscle cells functioning under physiological conditions. Since pyloric antrum propulsive activity is engaged in forming an increased pressure of food content necessary for appropriate stomach emptying [36], our neuroanatomical results imply the role of neuronal CART in the regulation of such process. However, immunohistochemical results presented by Zacharko et al. [37], and our earlier tracing study [38] clearly indicate that some of the CART-immunoreactive nerve fibers observed in the porcine gastric wall may originate in dorsal root ganglia and participate in the extrinsic sensory stomach innervation.

In healthy animals, singular, irregularly arranged submucosal CART-immunoreactive gastric perikarya were observed only in sporadic studies (present study; [12,32,39]) or were completely absent [11,31,34,40] suggesting their minor importance for intramural regulation of the stomach mucosa secretory function. The reasonable explanation for the abovementioned minor variations presented by different authors seem to result from tissue sampling differences, as described before.

Interestingly, in the other parts of gastrointestinal tract (in healthy animals), as an esophagus [33] and intestines [40,41], CART-immunoreactive submucosal neurons were encountered, although most of the CART-positive perikarya were also located in the myenteric plexus. Submucosal perikarya project mainly to the mucosa, other submucosal ganglia and submucosal blood vessels [42]. Obtained neuroanatomical results suggest minor importance of CART for the submucosal nerve regulation of gastric juice content under physiological conditions. However, CART modulatory function on the gastric acid secretion can occur via its influence on the central nervous system, as depicted by Okumura et al. [43].

The results of Q-PCR technique obtained in the present experiment demonstrated for the first time the presence of mRNA encoding CART in the stomach of healthy pigs, complementing the existing and only immunocytochemical data on its expression in the gastrointestinal tract of this species [11,12,31,34,40]. Since different mechanisms are engaged in the regulation of mRNA expression and final peptide abundance, the results of Q-PCR technique and immunohistochemistry may vary in the same tissues [44]. Therefore, the presented results are of particular importance, introducing another layer of research on the mechanism of CART expression regulation in the porcine gastrointestinal tract.

Pathological processes occurring within the gastrointestinal tube directly influence the structures of the intrinsic nerves [2]. Mucosa, as a layer directly bordering with the gastrointestinal lumen, is immediately exposed to ongoing pathological processes, suggesting strong reaction of submucosal neurons to pathology. Such reaction was observed in various gastrointestinal experiments [2,45,46,47,48]. Some gastrointestinal studies revealed the changes in the number of intestinal submucosal CART positive neurons during porcine proliferative enteropathy [49] as well as under experimental inflammation and axotomy [5]. Stomach is a part of gastrointestinal tract in which submucosal ganglia are strongly reduced, which, according to some authors [42], excludes the concept of submucosal plexus in this structure. Results obtained in the present study indicated a lack of CART involvement in the plasticity of submucosal gastric perikarya, which may result from the specificity of the studied stomach disorder. Gastric ulceration, which is a focal lesion penetrating deeper layers of the stomach wall, showed different galaninergic and tachyninergic reaction of submucosal neurons [16,17,18] in relation to other gastrointestinal inflammations affecting large areas of the mucosa. On the other hand, obtained results may be a consequence of an insignificant share of CART in the pyloric antrum submucosal neurons under physiological conditions, and subsequently, their minor importance in reaction to pathology. Surprisingly, significant increase of submucosal CART-immunoreactive perikaryal was observed in the stomach corpus of pigs experimentally treated with T-2 Fusarium toxin [12], which suggests a ‘toxic-dependent’ reaction of these nerve cells.

Our experiment revealed, for the first time, that stomach antrum ulcer evoked significant CART-dependent reaction of gastric myenteric neurons and nerve fibers localized within the stomach circular muscle layer. Since most of these fibers originate in myenteric perikarya [42], observed convergent reaction is logical and indicates significant decrease of CART-expression in enteric nerves supplying muscular layer. Similar reduction in CART-IF muscular layer neuronal fibers and myenteric perikarya was observed in the human stomach wall affected by tumor [39], and in pigs with experimentally induced diabetes mellitus [11]. Experimental intoxication of pigs with T-2 fusarium toxin resulted in CART-IF increase observed in all muscle layers of the porcine stomach wall [12]. Importantly, both above cited articles described multi-system swine pathologies which are not directly related with the stomach and cannot be classified as purely gastric disorders. Therefore, our results obtained in experimental stomach ulcers, a gastric disorder widely occurring around the world, are of particular significance. Interestingly, our results revealed, for the first time, changes in the characteristics of CART-immunofluorescent myenteric neurons under stomach pathology. Although the total number of CART-immunofluorescent myenteric neurons was reduced, their average size and fluorescence intensity were increased. This phenomenon indicates a complicated reaction of studied perikarya, in which the remaining group of neurons synthetize peptide with increased activity, possibly adjusting its release to precisely defined locations.

In the intestinal inflammatory studies, the number of CART-IF myenteric neurons was increased in ulcerative colitis in humans [9] and inflammatory bowel disease in dogs [40], while experimental colitis [5] and proliferative enteropathy [49] in pigs reduced the number of myenteric perikarya.

Our results showed that stomach myenteric perikarya reaction in ulcer disease is consistent to that observed in some intestinal inflammations and multi-system disorders examined in pigs. However, CART-IF neuronal fibers reaction revealed several discrepancies, which may result from a direct influence of deeply penetrating ulcers on the muscular layer.

Our Q-PCR analysis revealed, for the first time, a considerable decrease of mRNA encoding CART (relative expression to GAPDH) in the stomach tissues of ulcer animals. Such observation indicates that under fully developed acute ulcer disease (seven days after ulcer induction) the quantity of CART is still regulated at mRNA level. Its decreased expression is highly converged with reduced number of CART-immunofluorescent myenteric perikarya. On the other hand, changes in the myenteric perikarya fluorescence intensity (demonstrated in the present study), suggest post-translational precise regulation of CART synthesis in these cells.

The presented experiment revealed, for the first time, that CART is engaged in a specific reaction of the stomach enteric nervous system to gastric ulcerations. Our study completed the world literature in the first data on the CART-encoding gene expression in the swine gastrointestinal tract. The obtained neuroanatomical results complement the knowledge about the studied peptide and may contribute to a better understanding of its role in the stomach functioning in healthy individuals and under gastric ulceration—commonly occurring gastrointestinal pathology.

## 4. Materials and Methods

### 4.1. Animals and Experimental Procedures

All experimental protocols and handling of animals were approved by the Local Ethics Committee of the University of Warmia and Mazury in Olsztyn (permit number 76/2012) affiliated to the National Ethics Commission for animal experimentation (Polish Ministry of Science and Higher Education) and were performed in accordance with these guidelines. All efforts were made to minimize animals suffering at all stages of the experiment.

The study was performed on sexually immature female pigs (Polish Large White breed, body weight approximately 20 kg) obtained from a commercial fattening farm (14-260 Lubawa, Poland).

Animals (*n* = 24) were randomly assigned to control (C; *n* = 12) and experimental (U; *n* = 12) groups. In experimental animals, stomach antrum ulcers were evoked by bilateral submucosal injections of 1 cm^3^ of 40% acetic acid solution according to the acetic acid ulcer model procedure [50]. The procedure was performed under full anesthesia. Shortly: 30 min before the main anesthetic was given, animals were pre-treated with azaperone (Stresnil, Janssen Pharmaceutica, Beerse, Belgium, 0.4 mg/kg b.w., i.m.) and atropine (Polfa, Warsaw, Poland, 0.04 mg/kg b.w., s.c.); then, the animals were generally anaesthetized with xylazine (Vetaxyl, Vet-Agro, Lublin, Poland, 0.3 mg/kg b.w., i.m.) and ketamine (Bioketan, Vetoquinol Biowet, Gorzow Wielkopolski, Poland, 15 mg/kg b.w., i.v., qs), and the stomach was exposed via the midline laparotomy. Following acetic acid injection, a sterile tampon was sealed over the inserted needle for approximately 30 s to avid solution leakage and the accuracy of injection site was confirmed by a weal-like swelling in the place of injection. Finally, the abdomen incision was sutured and secured.

After surgery, the animals were moved to individual pens with unrestricted access to water (access to feed 24 h later) and kept for a week. Control animals were housed in the parallel pens and fed according to the same feeding procedure.

Since tissues for immunofluorescence and Q-PCR analyses require different fixation processes, the animals were divided into histochemical (H; control *n* = 6 and experimental *n* = 6; immunofluorescence staining; fixation by transcardial perfusion with 4% PFA solution) and molecular (M; control *n* = 6 and experimental *n* = 6; Q-PCR; fixation in RNA-later) subgroups at the terminal stage.

### 4.2. Tissue Preparation

Stomachs collected from all the pigs were cut along the greater curvature and thoroughly washed in PBS to remove food debris. Gastric wall tissue samples adjacent to the ulcer (stomach antrum, glandular part of the organ) were taken in experimental animals, while in control animals, similar samples at sites strictly corresponding to those sampled in experimental pigs were obtained. To ensure the correspondence of the tissue samples collected from each animal, the distance from the pyloric orifice was precisely measured in each animal studied.

Afterwards: H-group samples were post–fixed in 4% PFA solution (60 min), rinsed in PBS, immersed in 18% buffered (pH 7.4) sucrose solution, cut into 20 µm thick transverse cryostat consecutive tissue sections, mounted on chrome alum–gelatine–coated slides, air–dried and stored desiccated at −23 °C; M-group samples were immersed in 4 °C RNAlater^®^ (Ambion, Austin, TX, USA) overnight and stored at −80 °C.

### 4.3. Double-Labeling Immunofluorescence

Double immunofluorescence staining was performed on tissue sections collected from H-group of animals. The tissue slides were processed for routine double immunofluorescence staining with a mixture of primary antibodies against pan-neuronal marker PGP 9.5 (mouse anti-PGP 9.5, dilution 1:600, code 7863–2004, clone 31A3, AbD Serotec, Raleigh, NC, USA) and CART (rabbit anti-CART, dilution 1:20,000, code 8450-0505, AbD Serotec), and corresponding secondary antibodies (AlexaFluor 488, goat anti-mouse, dilution 1:500, code A11001 and AlexaFluor 555, goat anti-rabbit, dilution 1:500, code A-21428, Invitrogen, Waltham, MA, USA). The primary antibodies applied in the study were recommended for application in the porcine tissues. All staining procedures and controls applied in the experiment were precisely described in the previous article [51] and confirmed the specificity of the procedure.

Microscopic analysis was performed under the confocal microscope (LSM700, Zeiss, Jena, Germany) equipped with lasers and filter sets for AlexaFluor488 and AlexaFluor555. Analyzed tissue sections were separated by a distance of at least 80 µm (greater than dimensions of the largest intramural perikarya), ensuring that none of the immunofluorescent perikaryon was counted twice. The slides were evaluated by a team of two investigators (the same persons), each analyzing the same number of sections in each animal studied. During the analysis, the researcher was blinded to the experimental group—tissue slides were identified by laboratory technicians and announced to the investigator only after finishing analysis. At least 700 of PGP 9.5-positive cell bodies in the myenteric plexus and 200 submucosal perikarya were analyzed in each studied animal. To determine the percentages of CART-immunoreactive myenteric cells, the number of neurons simultaneously co-expressing pan-neuronal marker PGP 9.5 and CART were counted. The results were presented as average percentages ± SEM for control and experimental groups. Due to the irregular and only occasional occurrence of submucosal CART-immunoreactive perikarya in just a few individuals of the control and experimental groups, these results were not statistically analyzed, and were described in the results and presented on photomicrographs.

ImageJ analyses were performed on confocal laser microscope photomicrographs (without z-stack resolution) captured with identical parameters for each type of analysis. The intensity of CART-immunofluorescence expressed by myenteric perikarya was determined on a group of at least 20 PGP 9.5/CART–double immunoreactive randomly selected perikarya (on microphotographs) in each animal studied using ImageJ analysis (mean grey value measured inside the perikaryon area) and presented in optical density units (u).

Myenteric perikarya dimensions (longitudinal and transverse) were measured with confocal laser microscope software on a group of at least 20 perikarya with visible nucleus in each animal studied.

Because of the huge variability in the longitudinal muscle layer thickness existing between different parts of the same microscopic slide (collected from the same animal), and due to the highly irregular occurrence of CART-immunofluorescent nerve fibers in the submucosa and lamina muscularis mucosae, the densities of CART-immunofluorescent nerve fibers located in these layers were evaluated arbitrary by the observers.

CART-immunofluorescent nerve fibers in the circular muscular layer (distributed more evenly throughout the microscopic section) were quantitatively analyzed on at least 15 confocal captured microphotographs taken from predetermined areas in each animal studied. The CART-immunofluorescence expressed by nerve fibers was analyzed with ImageJ software (“Intermodes” threshold algorithm applied) and determined as a percentage of the surface area occupied by CART-fluorescent fibers (in the field of view).

Finally, in all the quantitative analyses, the mean ± SEM was calculated for control and experimental groups and further analyzed for statistically significance (as described below).

Photo-documentation has been prepared using a confocal laser microscope (LSM700, Zeiss, Jena, Germany)). Set of figures and drawings was prepared with CorelDRAW X7 ver. 17.6.0.1021 (Ottawa, ON, Canada) graphical software.

### 4.4. Real-Time PCR

The real-time PCR was applied on samples collected from the M-group animals. Total RNA was isolated from 50 µg of tissue homogenate (perpendicular section covering all the stomach wall layers) using a Total RNA Mini Plus kit (A&A Biotechnology, Gdansk, Poland).

Maxima First Strand cDNA Synthesis Kit for RT-qPCR (code K1672, Thermo Fisher Scientific, Waltham, MA, USA) and 1.5 μg of total RNA were used to produce cDNA. Primers for porcine CART (Forward: TATGTGTGACGCAGGAGAGC; Reverse: AAGGAATTGCAGGAGGTTCC) were described by Li et al. [52] and validated with Primer-BLAST online software (http://ncbi.nlm.nih.gov). Previous experiment [16] had indicated GAPDH as a relevant and reliable housekeeping gene for the study (with stably expressed values for all the animals), and previously designed primers (Forward: TTCCACCCACGGCAAGTT; Reverse: GGCCTTTCCATTGATGACAAG) were used in the study.

The PCR reaction was performed in 7500 fast real-time PCR system (Applied Biosystems, Waltham, MA, USA) with the thermal profile consisting of: initial denaturation 10 min at 95 °C, denaturation 15 s at 95 °C, and annealing 1 min at 60 °C for 40 cycles. CART expression data were normalized to GAPDH using software 7500 v. 2.0.2 (Applied Biosystems, Waltham, MA, USA). cDNA samples were amplified in doublets and data were expressed as the mean ± SEM (for control and experimental groups).

### 4.5. Statistical Analysis

The differences in the quantitative results obtained from the control and experimental animals were statistically analyzed with GraphPad Prism Software Inc., San Diego, CA, USA, ver. 6.05 for Windows. Firstly, the D’Agostino and Pearson omnibus normality test was performed. Then, Student’s *t*-test (normal distributed data), Mann–Whitney U test (for non-normal distributed data) or one way ANOVA followed by Bonferroni’s post hoc test (for multiple comparisons of cell dimensions) were performed and obtained results were considered to be significant at *p* < 0.05.

## Figures and Tables

**Figure 1 ijms-22-07437-f001:**
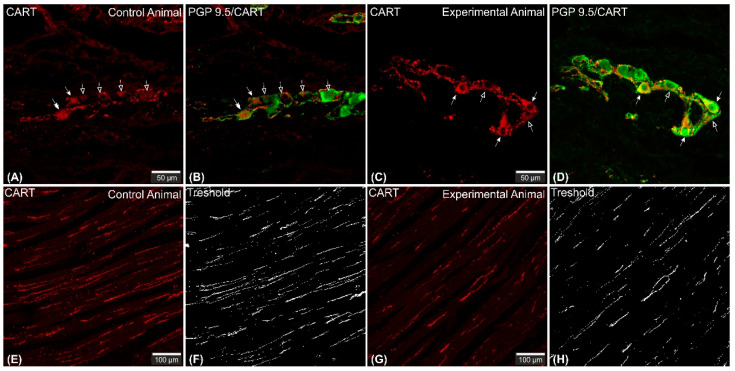
Set of exemplary photomicrographs showing myenteric plexus perikarya (**A**–**D**; arrows) and nerve fibers (**E**,**G**) penetrating the circular muscle layer in the control (**A**,**B**,**E**,**F**) and experimental animal (**C**,**D**,**G**,**H**) immunoassayed with antibodies against PGP 9.5 (green channel) and CART (red channel). The number of PGP/CART-IF double immunostained perikarya in experimental animals (**C**,**D**) was lower than in control pigs (**A**,**B**). Solid arrows point to highly CART-fluorescent perikarya, empty arrows point to weakly fluorescent cell bodies. In experimental animals perikarya were characterized by stronger immunofluorescence (**C**) than in control ones (**A**). (**E**,**G**) clearly visible CART-IF varicose nerve fibers penetrating circular muscle layer. (**F**,**H**) Intermodes threshold method (ImageJ software ver. 1.53c, Rasband, W.S., National Institutes of Health, Bethesda, MD, USA) applied on red channel (CART immunostaining) microphotographs enabled precise quantitative analysis of CART-immunofluorescent nerve fibers in the circular muscle layer indicating their reduction in ulcer animals. Scale bars presented in the pictures.

**Figure 2 ijms-22-07437-f002:**
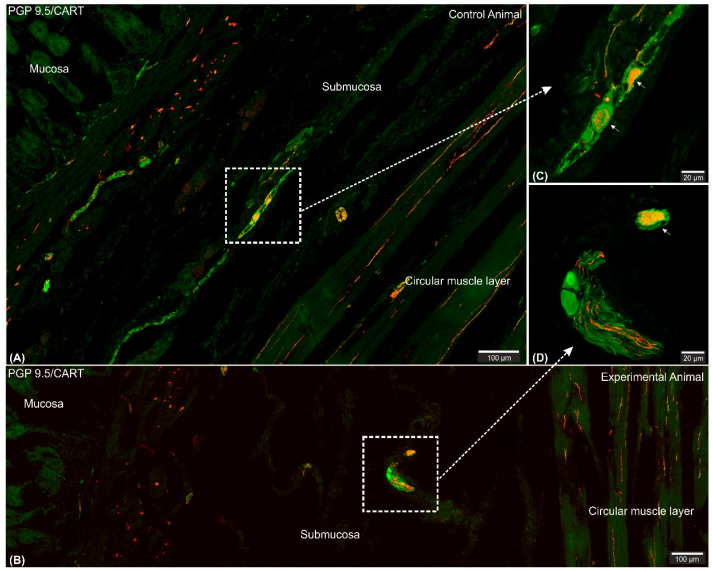
Set of exemplary figures showing submucosal PGP 9.5/CART double immunofluorescent perikarya (arrows) sporadically observed in the control (**A**,**C**) and experimental (**B**,**D**) pigs. (**A**,**B**) Figures showing large area of the stomach wall (mucosa, submucosa, and deeper part of the circular muscle layer) created using the confocal ‘tilt’ function to visualize the exact location of submucosal perikarya. (**B**,**D**) Higher magnification of submucosal perikarya. Both channels (red—CART staining, green—PGP 9.5 staining) overlapped in the pictures. Scale bars included in the pictures.

**Figure 3 ijms-22-07437-f003:**
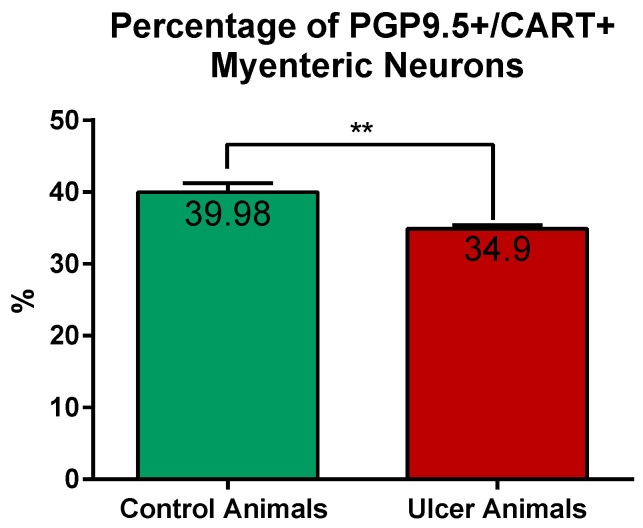
Graph showing the percentage of PGP 9.5/CART double immunofluorescent myenteric perikarya in the control and experimental animals. The number of cells was significantly reduced in ulcer animals. (**) *p* value = 0.0087.

**Figure 4 ijms-22-07437-f004:**
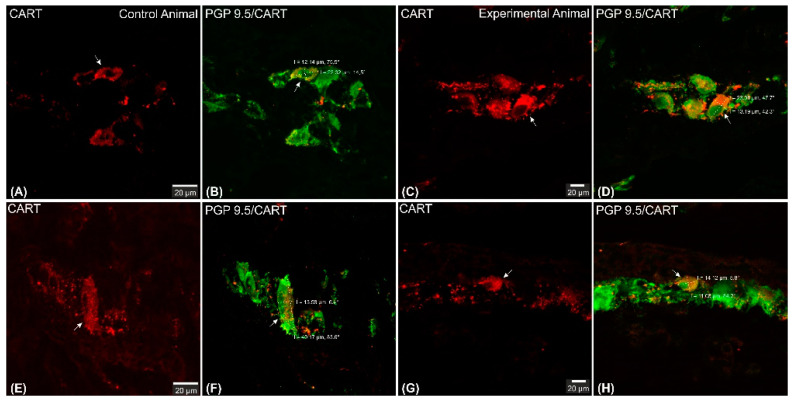
Set of exemplary photomicrographs double immunostained for CART (red channel) and PGP 9.5 (green channel) showing morphology (including dimensions) of double immunofluorescent myenteric perikarya (arrows). (**A**–**D**)—pictures indicating typical myenteric perikaryon (arrow) observed in the control (**A**,**B**) and experimental (**C**,**D**) animals. High intensity of CART-immunofluorescence observed in the experimental animal perikaryon (**C**). (**E**,**F**) Very large and (**G**,**H**) small perikarya occasionally observed in both groups of animals. Scale bars included in the pictures. Both channels overlapped in (**B**,**D**,**F**,**H**) pictures. Measurements (long/short axis) made with confocal microscope software and incorporated by graphical software.

**Figure 5 ijms-22-07437-f005:**
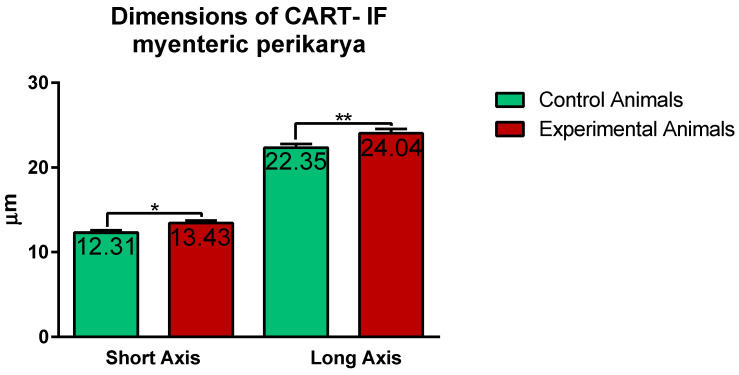
Graph showing the average dimensions (long and short axis) of myenteric perikarya in the control and experimental animals. Both dimensions were increased in ulcer animals. (*) *p* value = 0.0226; (**) *p* value = 0.0044.

**Figure 6 ijms-22-07437-f006:**
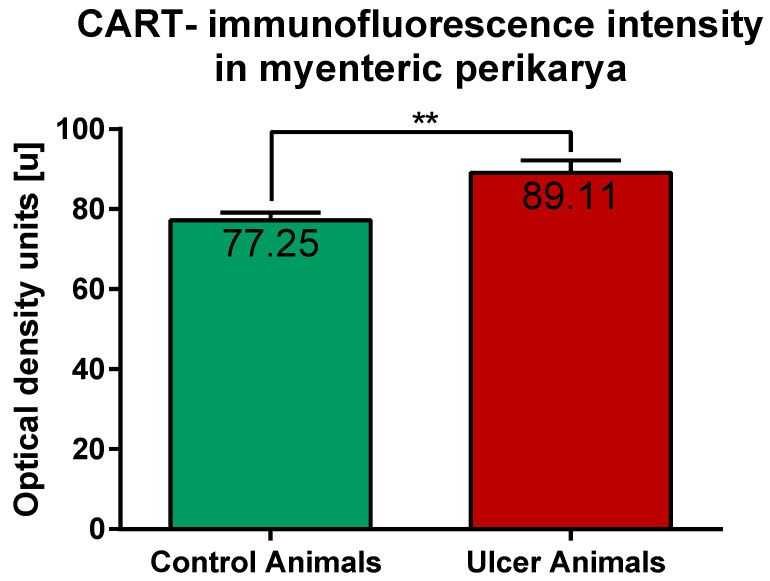
Graph showing the average intensity of CART-immunofluorescence expressed by control and experimental pigs’ myenteric perikarya measured with ImageJ software (ver. 1.53c, Rasband, W.S., National Institutes of Health, Bethesda, MD, USA) as a mean grey value inside the cell body area. The values are presented in optical density units. In ulcer animals the average intensity was higher than in control pigs. (**) *p* value = 0.006.

**Figure 7 ijms-22-07437-f007:**
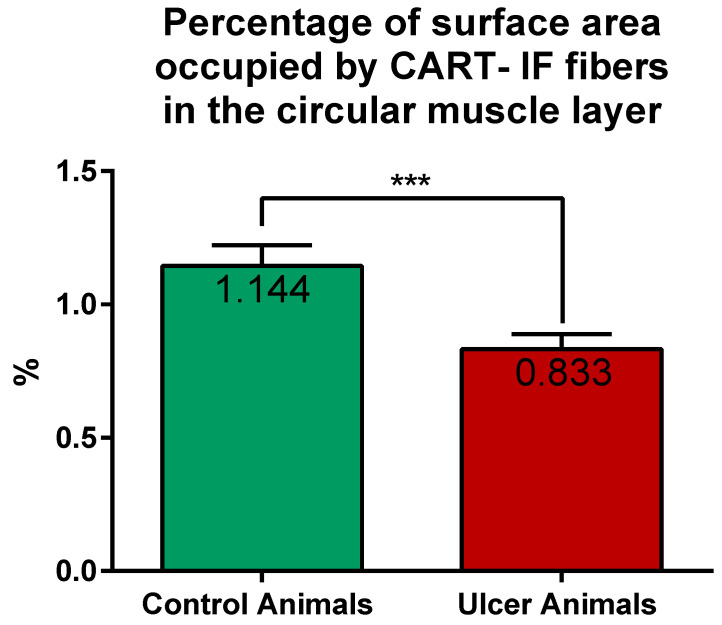
Graph showing the percentage of surface area (in the field of view) occupied by CART-immunofluorescent nerve fibers located in the circular muscle layer in the control and ulcer pigs. The value decreased in animals with ulcers. (***) *p* value = 0.0006.

**Figure 8 ijms-22-07437-f008:**
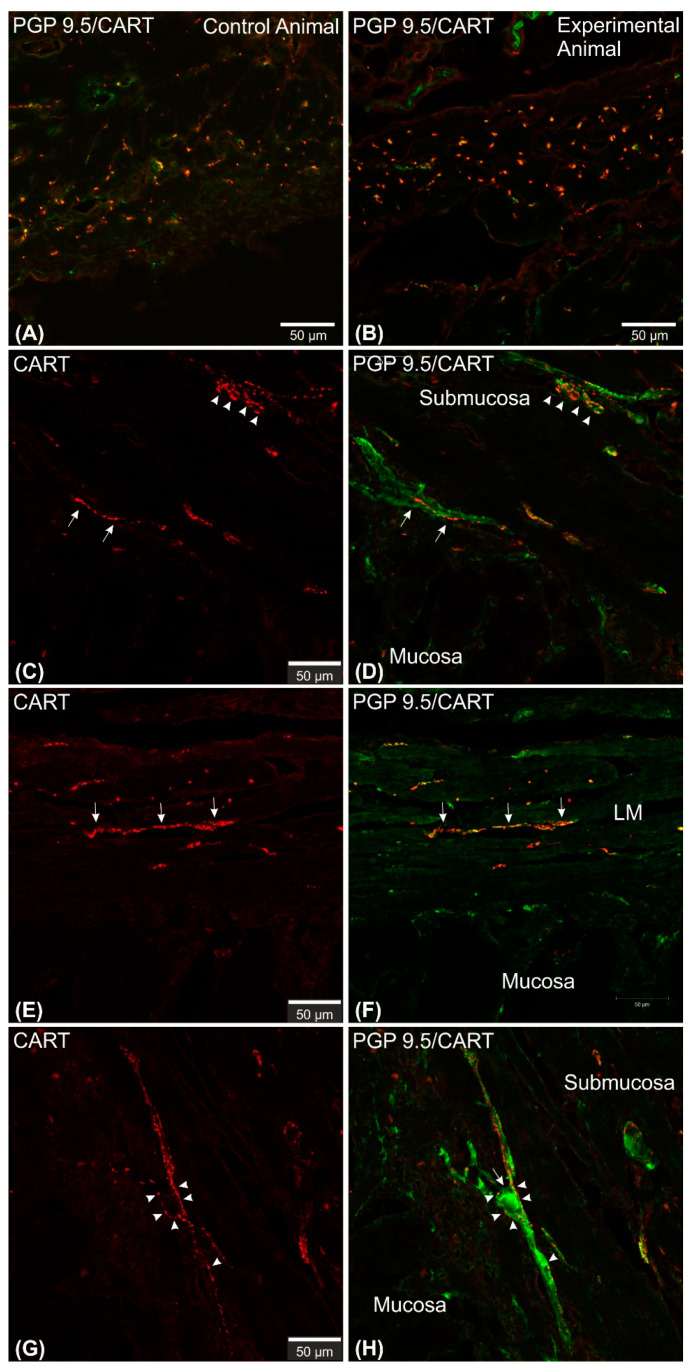
Set of exemplary microphotographs showing CART/PGP 9.5 immunolabelled neuronal fibers in the longitudinal muscle layer (**A**,**B**) and in the submucosa and lamina muscularis mucosae (**C**–**H**). (**A**,**B**) Evenly dispersed, transversally cut neuronal fibers observed in the control (**A**) and experimental (**B**) animals. (**C**,**D**) Bundles of highly fluorescent varicose nerve fibers located within submucosa (arrowheads) and sparse nerve fibers observed in the lamina muscularis mucosae (arrows) in both studied animal groups. (**E**,**F**) Thick, highly immunoreactive fibers running parallel to the muscle fibers of muscularis mucosae (arrows) sporadically observed in both groups of animals. (**G**,**H**) CART-immunofluorescent nerve fibers (arrowheads) attaching and encircling perikaryon located in the lamina muscularis mucosae (arrow) observed in both studied groups of pigs. Scale bars included in the pictures. CART—red channel, PGP 9.5—green channel. In (**A**,**B**,**D**,**F**,**H**) both channels overlapped. LM—lamina muscularis mucosae.

**Figure 9 ijms-22-07437-f009:**
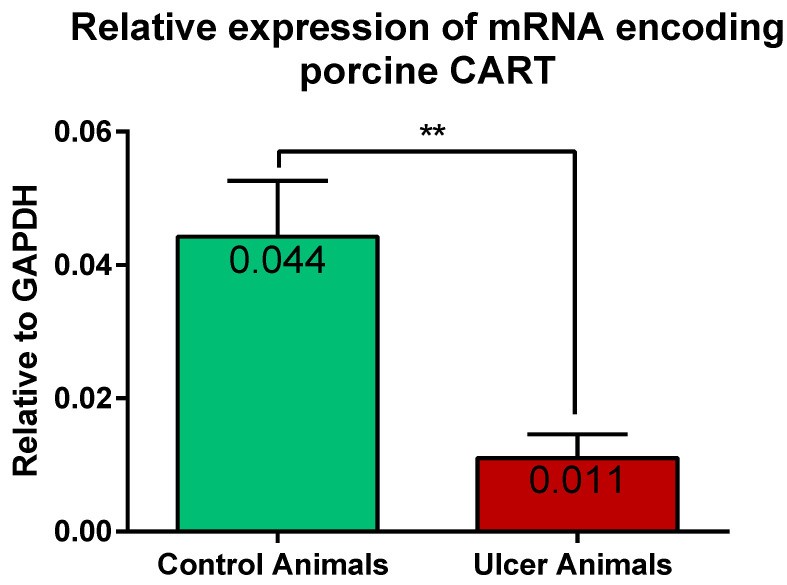
Graph showing the expression level of mRNA encoding CART in the stomach antrum tissues of the control and ulcer pigs. The stomach antrum circular-shaped samples (directly adjacent to the ulcer verge in experimental animals; at the strictly equivalent location in control animals) were cut transversally to the gastric wall and comprised all its layers. The data obtained from each sample were normalized to GAPDH. Relative quantities (RQ) of mRNA were analyzed using the comparative Ct method. (**) *p* value = 0.0043.

## Data Availability

All relevant data are within the manuscript.

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
