# Peer review of "Cocaine and Amphetamine Regulated Transcript (CART) Expression Changes in the Stomach Wall Affected by Experimentally Induced Gastric Ulcerations"

_ijms, 2021, doi:10.3390/ijms22147437_

Round 1
Reviewer 1 Report
Gastric ulceration has been a clinical issue in human and pig, thus pathophysiological examination and biomarker discovery are both important for the development of treatment. Zalecki M et al first characterized a porcine model of gastric ulcer by immunofluorescence technique. Then, these authors demonstrated the impaired expression of CART mRNA by qPCR.
Minor comments
1) In Fig 9 legend, the sampling position of stomach should be described.
2) It would be interesting to discuss serum CART levels whether this could be a biomarker for gastric ulceration in this model.
3) "Discussion" section seemed too long.
Author Response
The authors would like to kindly thank the reviewer for evaluating the manuscript and all the valuable comments. All efforts were made to fulfill all the reviewer’s expectations.
We have referred to all the suggestions raised by the reviewer:
1. In Fig 9 legend, the sampling position of stomach should be described.
We would like to kindly inform, that the precise description of the sampling position was incorporated into Fig 9 legend (lines 224 – 227). Moreover, the P value was added.
2. It would be interesting to discuss serum CART levels whether this could be a biomarker for gastric ulceration in this model.
The conception presented by the Reviewer seems to be interesting, especially in terms of practical application. Unfortunately, the source, fate and regulatory mechanisms of blood circulating CART peptide are poorly understood. The literature focused on the blood level of CART is very scarce and no studies relate to gastrointestinal disorders. Regrettably, our study does not allow for the formulation of a certain hypothesis that could be incorporated into the article.
However, based on the available literature we can speculate that plasma levels of CART might be elevated in gastric ulcer patients, and these changes are highly probably related with the changes in glucocorticoids. For several decades it is generally accepted that glucocorticoids are highly associated with gastric ulcerations, however their pro- or anti-ulcer role is being extensively discussed (doi: 10.1177/2040622311412420). As it was depicted by Vicentic et all (DOI: 10.1210/en.2003-1648) the levels of CART peptides in the blood are highly (in 70%) regulated by corticosterone and may be influenced by hypothalamic-pituitary-adrenal interactions. In line, the lipopolysaccharide‑induced systemic inflammation (doi: 10.1186/s12868-019-0539-z) and pro-inflammatory tumor model (DOI: 10.3892/or.2016.4558) revealed increased level of plasma CART, although its levels in certain tissues (e.g., brain regions) were significantly reduced (as in our “gastric ulcer” experiment). Presented conception, although purely hypothetical, seems to be supported by knowledge gained on CART's role in controlling appetite and energy homeostasis ( DOI: 10.3389/fnins.2014.00313) – factors important in various diseases, including gastric ulcers.
3. "Discussion" section seemed too long.
We kindly inform that the section was shortened (lines 282-283; 291-293; 312-314, 335-338, 354-359 were removed).
Reviewer 2 Report
The manuscript seems interesting and may constitute the basis for further clinical applications. It is generally well written and includes useful statistical analysis. Figures are of good quality.
Minor comments:
- It needs to be specified which precisely part of the stomach was dissected out for the study: nonglandular or glandular (cardia, corpus or antrum region)? It is extremely important because most of the available studies report the occurrence of ulcers predominantly in the non-glandular region whereas only sparse ulcers were reported in glandular portion of the porcine stomach.
- It has been also reported that CART is expressed in the porcine dorsal root ganglia (See Zacharko-Siembida et al., 2014 Co-expression patterns of cocaine- and amphetamine-regulated transcript (CART) with neuropeptides in dorsal root ganglia of the pig. Acta Histochemica 116:390-398). Because it is well known that DRGs are the source of primary afferent neurons supplying the gut, it is possible that at least some of CART-IR nerve fibres observed by the authors in the stomach wall are sensory in nature. This work/issue should be included in the study and discussed.
- The authors should determine how many persons were involved in neuronal counting/immunoreactivity assessment procedures. Did they average the obtained data somehow?
- What was post-hoc test used with ANOVA ?
- Please correct the reference 44 (names)
Author Response
The authors would like to kindly thank the reviewer for the evaluation of the manuscript and for all the valuable comments. All efforts were made to fulfill all the reviewer’s expectations.
We have referred to all the suggestions raised by the reviewer:
1. It needs to be specified which precisely part of the stomach was dissected out for the study: nonglandular or glandular (cardia, corpus or antrum region)? It is extremely important because most of the available studies report the occurrence of ulcers predominantly in the non-glandular region whereas only sparse ulcers were reported in glandular portion of the porcine stomach.
We would like to kindly inform that the ulcers were experimentally induced in the stomach antrum, and this glandular part was dissected out for the study. Although there are reports on different localization of ulcers (some of which indicate a non-glandular portion of the stomach as a predominant area for porcine ulcers) we decided to investigate the main region responsible for control of the gastric emptying process. It is known that pyloric antrum propulsive activity is engaged in forming an increased pressure of food content necessary for appropriate gastric emptying (doi:10.1136/gut.4.2.174). Importantly, precisely regulated outflow of the gastric content is extremely important for the homeostasis of the whole organism - maldigestion, malabsorption and malnutrition are the consequences of disrupted gastric emptying process. The impaired gastric motility in the distal stomach and problems with gastric emptying were described in patients with gastrointestinal ulcerations since the forties of the last century (doi.org/10.1007/BF02996942, DOI: 10.1007/BF02233838). Gastric emptying is disturbed, especially when the ulcer is located in the distal part of the stomach (pyloric antrum), while proximal gastric or duodenal ulcer locations accelerate gastric outflow. Such specific symptoms have been even described as a “pyloric syndrome complex” already in the sixties of the last century (DOI: 10.1016/0002-9610(67)90222-x ).
Relevant text was incorporated into: Introduction (lines 60-62; 71), Discussion (line 328), Materials and Methods (391, 411-412), figure legend (lines 224 – 227). Reference list was complemented with appropriate literature.
We sincerely hope that the above arguments convince the reviewer of the advisability of examining this part of the stomach.
2. It has been also reported that CART is expressed in the porcine dorsal root ganglia (See Zacharko-Siembida et al., 2014 Co-expression patterns of cocaine- and amphetamine-regulated transcript (CART) with neuropeptides in dorsal root ganglia of the pig. Acta Histochemica 116:390-398). Because it is well known that DRGs are the source of primary afferent neurons supplying the gut, it is possible that at least some of CART-IR nerve fibres observed by the authors in the stomach wall are sensory in nature. This work/issue should be included in the study and discussed.
We are grateful to the reviewer for this comment. We totally agree that primary afferent neurons supply gastrointestinal tract, and even our previous tracing study showed exact location of DRG neurons innervating the part of the porcine stomach - pylorus (doi: 10.1007/s12031-013-0116-3). Therefore, some of the CART-immunofluorescent nerve fibers observed in the stomach wall may originate in DRGs and participate in the extrinsic sensory stomach innervation.
The appropriate text was incorporated into Discussion (lines 276-280) and the list of references was supplemented with the required publications.
3. The authors should determine how many persons were involved in neuronal counting/immunoreactivity assessment procedures. Did they average the obtained data somehow?
We kindly inform, that slides were microscopically analyzed by a team of two observers (the same persons for all the animals studied). Each investigator analyzed the same number of sections in each animal studied. Then the results were combined, and the percentage of PGP 9.5 / CART double immunoreactive myenteric perikarya was calculated for each animal studied.
The relevant information was included into the manuscript (4.3. Double-Labelling Immunofluorescence, lines 435-437)
For software analysis (ImageJ, confocal software analysis) one researcher did the analyses.
What was post-hoc test used with ANOVA ?
We would like to inform, that Bonferroni’s post hoc test was used with ANOVA. Appropriate information was incorporated into lines: 490 – 491.
5. Please correct the reference 44 (names)
We kindly inform that the reference was corrected (in the improved version of the article the reference number is 49). Moreover, other minor corrections were incorporated in selected citations (e.g. position 33, 47).
Round 2
Reviewer 1 Report
Dear the Editor
The authors submitted the revised manuscript that reasonably responded to all queries raised by the Reviewer.